# Plasma Lipidomic Profiling Using Mass Spectrometry for Multiple Sclerosis Diagnosis and Disease Activity Stratification (LipidMS)

**DOI:** 10.3390/ijms25052483

**Published:** 2024-02-20

**Authors:** Seyed Siyawasch Justus Lattau, Lisa-Marie Borsch, Kristina auf dem Brinke, Christian Klose, Liza Vinhoven, Manuel Nietert, Dirk Fitzner

**Affiliations:** 1Department of Neurology, University Medical Center Göttingen, 37075 Göttingen, Germany; justus.lattau@med.uni-goettingen.de (S.S.J.L.); lisa-marie.borsch@med.uni-goettingen.de (L.-M.B.); kristina.brinke@med.uni-goettingen.de (K.a.d.B.); 2Lipotype GmbH, 01307 Dresden, Germany; klose@lipotype.com; 3Department of Medical Bioinformatics, University Medical Center Göttingen, 37075 Göttingen, Germany; liza.vinhoven@med.uni-goettingen.de (L.V.); manuel.nietert@med.uni-goettingen.de (M.N.)

**Keywords:** multiple sclerosis, lipidmetabolism, lipidomics, biomarker

## Abstract

This investigation explores the potential of plasma lipidomic signatures for aiding in the diagnosis of Multiple Sclerosis (MS) and evaluating the clinical course and disease activity of diseased patients. Plasma samples from 60 patients with MS (PwMS) were clinically stratified to either a relapsing-remitting (RRMS) or a chronic progressive MS course and 60 age-matched controls were analyzed using state-of-the-art direct infusion quantitative shotgun lipidomics. To account for potential confounders, data were filtered for age and BMI correlations. The statistical analysis employed supervised and unsupervised multivariate data analysis techniques, including a principal component analysis (PCA), a partial least squares discriminant analysis (oPLS-DA) and a random forest (RF). To determine whether the significant absolute differences in the lipid subspecies have a relevant effect on the overall composition of the respective lipid classes, we introduce a class composition visualization (CCV). We identified 670 lipids across 16 classes. PwMS showed a significant increase in diacylglycerols (DAG), with DAG 16:0;0_18:1;0 being proven to be the lipid with the highest predictive ability for MS as determined by RF. The alterations in the phosphatidylethanolamines (PE) were mainly linked to RRMS while the alterations in the ether-bound PEs (PE O-) were found in chronic progressive MS. The amount of CE species was reduced in the CPMS cohort whereas TAG species were reduced in the RRMS patients, both lipid classes being relevant in lipid storage. Combining the above mentioned data analyses, distinct lipidomic signatures were isolated and shown to be correlated with clinical phenotypes. Our study suggests that specific plasma lipid profiles are not merely associated with the diagnosis of MS but instead point toward distinct clinical features in the individual patient paving the way for personalized therapy and an enhanced understanding of MS pathology.

## 1. Introduction

Multiple sclerosis (MS) is one of the most common inflammatory demyelinating diseases of the central nervous system (CNS). With its rising prevalence, MS is viewed as a significant cause of neurological disability in adult life [1]. Previous metabolic studies in patients with MS (PwMS) have demonstrated the relevance of serum lipid composition not only as a potential diagnostic biomarker but also as a key to pathophysiological understanding [2]. Notably, postmortem brain tissue analyses from PwMS have also revealed distinctive lipid alterations in normal-appearing white matter [3]. Subsequently, a lipidomic study showed differences in the plasma of monozygotic twins discordant for MS [4]. These studies revealed alterations in the composition of the ether phosphatidylethanolamines (PE O-) and ether phosphatidylcholines (PC O-) between PwMS and controls neglecting individual disease courses. Therefore, it remains unclear if alterations in lipid profiles can indicate clinical activity or states of the disease. The prediction and evaluation of disease activity is crucial for tailoring individualized therapeutic approaches [5]. Thus, readily available biomarkers are required not only to aid in the diagnosis and monitoring of disease activity as well as treatment responses, but also to provide insight into pathophysiological mechanisms [5].

In recent years, various methods with different performances have been proposed, such as serum neurofilament [6,7], an assessment of the retinal nerve fiber layer [8], intrathecal immunoglobulin M synthesis [9,10], serum glial fibrillary acid protein (GFAP) [11,12], a kappa free light chains (KFLC) index and KFLC intrathecal fraction [13] as well as MRI-based biomarkers [14]. Given the prominent destruction of the lipid-rich myelin sheath in MS, the analysis of lipid composition has revealed itself as a promising approach [2,4,15,16]. Our study uses a state-of-the-art direct infusion (shotgun) nano-electrospray high-resolution Orbitrap mass spectrometry [17,18] to analyze plasma samples from 120 participants, including 30 patients with relapsing-remitting MS (RRMS), 30 patients with chronic progressive disease courses of MS and an age-matched control group of 60 individuals.

## 2. Results

### 2.1. Lipid Class Variations across Cohorts

A mass spectrometry analysis of serum samples was performed for 120 individuals and revealed 670 distinct lipids in 16 separate classes.

The dimensionality reduction analysis performed via PCA (Figure 1A) and LipidSpace (Appendix A) indicates a high similarity between the lipid patterns observed among the four cohorts. To avoid the overrepresentation of lipids with high correlations to age or BMI, a correlation analysis for age and BMI within the two control cohorts was conducted (Appendix A). Since lipids can be categorized into distinct lipid classes according to their molecular features, the first subsequent analysis focused on investigating whether lipid amounts differ within these classes and might aid in distinguishing cohorts. The ANOVA analysis with a Tukey post hoc test showed significant changes in the cholesteryl ester (CE), diacylglycerol (DAG), phosphatidylinositol (PI), sphingomyelin (SM), ceramide (Cer), phosphatidylethanolamine (PE) and ether-linked phosphatidylethanolamine (PE O-) classes (Figure 1B and Appendix A). Significant decreases in PE, PE-O, and PI were detected in the two MS cohorts. A reversed phenomenon was observed for DAG, with a significant increase in the RRMS and CPMS cohorts compared to the healthy cohort. The SM and CE classes showed a uniform pattern of increase in CPMS and OND. In addition, we performed a statistical evaluation of differences in the quantity of the individual fatty acid chains based on the chain length in relation to the respective lipid class. A significant difference between RRMS and CPMS was found only for the C22 chain of PC O- (Appendix A).

### 2.2. Comparative Analysis of Lipid Subspecies Variation in MS Cohorts

In the first step of the comparative analysis, we compared the combined cohorts of healthy and OND (non-MS) with the RRMS and CPMS (MS). The result was visualized using a volcano plot, modified by the weighting from a supervised machine oPLS-DA (Appendix A) in addition to an ordinary Welch’s *t*-test. The quality of the oPLS-DA was evaluated by using a test dataset, applying the model to predict the class (Appendix A), and assessing the R2X and R2Y values (Appendix A). A decrease in a subset of lipid species of the classes PC, PC O-, PE, PE O-, and PI, in contrast to an increase in DAG and selected PC subspecies, was found in the MS cohort (Figure 2). This statistical observation was supported by a heatmap with hierarchical clustering (Appendix A).

Subsequently, cohorts were analyzed separately and evaluated in an equal manner. Results are displayed in Figure 3. The age-matched direct comparison of the healthy cohort with the RRMS course is shown in Figure 3A. The importance of the PC- and PE subspecies composition in differentiating the two cohorts is supported by the VIPs of the corresponding oPLS-DA (Figure 3B). The predictive power of this model is confirmed by the confusion matrix in Figure 3C. Figure 3D contrasts OND with CPMS and emphasizes the alterations in PE O- subspecies. This finding is supported by the oPLS-DA, which shows the discriminatory potential of lipid subspecies (Figure 3E). The predictive power of this discrimination was tested on a holdout dataset. The result shows precise predictive power (Figure 3F).

The common pattern emerging from the pooled (Figure 2) and separated cohort analysis (Figure 3) of the two MS cohorts is characterized by a reduction in the PC, PE (more prominent in RRMS, Figure 3A,B) and PE O- (more prominent in CPMS, Figure 3D,E) and an elevation in the DAG, which is more pronounced in RRMS but observed in both RRMS and CPMS (Figure 3A,B). Notably, there are some overlapping lipid species, such as DAG 16:0;0_18:1;0 and DAG 18:0;0_18:1;0 with a large log_2_ fold change (log_2_FC) and low *p*-value (Figure 3A,D). A Venn diagram (Figure 4A) visualizes the overlap of the significantly altered lipids from the comparison in Figure 2 and Figure 3A,C, as well as the RF data detailed in Figure 5A. The overlap between the RRMS and CPMS cohorts shows an intersection of 67 lipids (42% of all significant changes in RRMS and 52% of all significant changes in CPMS). This points toward the existence of shared alterations of the plasma lipid composition in both, RRMS and CPMS patients, possibly indicating distinct plasma lipid profiles in PwMS. Furthermore, the link between significant differences in the lipid composition and clinically defined cohorts of RRMS and CPMS, as shown by a notable number of distinct lipid species altered, becomes apparent in the Venn diagram (Figure 4A). Using the Venn diagram, the significantly altered lipids were assigned to each cohort (RRMS, CPMS and MS cohort) and termed ‘cohort-specific lipid signatures’ or ‘profiles’ (Figure 4B–E).

### 2.3. Random Forest Validates Lipid Pattern for MS Diagnosis

To further validate the patterns discovered by the volcano plot and oPLS-DA (Figure 2 and Figure 3A,C) and explore the predictive capability, we employed an RF model to classify MS and non-MS samples. RF, as a non-linear machine learning technique, offers the ability to capture complex patterns that linear methods, like oPLS-DA, might not fully elucidate [19]. Therefore, previous studies in lipidomics have shown that it is advisable to use multiple machine learning approaches [19,20,21]. To generate a more parsimonious model, lipids with high correlations to BMI and age were removed and a multicollinearity filter was applied to further eliminate lipids with substantial inter-correlations. The model was trained on 60% of the data, tested via a 5-fold cross-validation and predicted the remaining 40% of the data. After 20 iterations, the model achieved an area under the curve (AUC) of 1 in the ROC for discriminating between MS and non-MS samples (Appendix A). Therefore, across all iterations, the model correctly classified all 24 MS patients (Figure 5B). However, these findings, while encouraging, warrant caution due to potential overfitting and the need for further validation with a larger, independent cohort. Interestingly, 25 out of the 30 most important lipids from the RF (Figure 5A) were revealed to be also significantly altered in the volcano/oPLS-DA approach, proving that the patterns presented here (Figure 4B–E) possess high specificity and sensitivity across 120 participants and thereby accurately reflect a distinct plasma lipid profile in PwMS. This demonstrates the predictive power of a lipid pattern as a diagnostic tool.

To determine whether the cohort-specific lipid signatures (Figure 4B–E) have a relevant effect on the overall composition of their lipid class, we introduce the class composition visualization (CCV). In this approach, we calculated the total quantity of the lipid classes in each of the four cohorts and visualized the cohort-specific lipid signatures (Figure 4B–E) with respect to the corresponding lipid class to evaluate the influence of the lipid profile on the class composition (Figure 5C).

### 2.4. Divergent Trends between Ether and Non-Ether Variants

In detail, the most frequent changes were detected in the PC species (Appendix A), which comprise up to 25 to 30% of all significant changes. Aside from an increase in PC 18:0;0_18:0;0 in the MS profile (Figure 4B,C) and a slight increase in PC 17:0;0_18:3;0 in the CPMS profile (Figure 4D), there was a marked reduction most dominant in the RRMS cohort (Figure 4B). Notably, the 24 significant lipids altered in both RRMS and CPMS made up less than 1% of the total abundance of PC in all four cohorts (Figure 5C). In contrast, the PC subspecies attributed to RRMS or CPMS accounted for approximately 20% of the total mass (Figure 5C).

The second most common characteristic of the here-presented MS profile is a decrease in the PC O- subspecies (Figure 4B,C and Appendix A). These changes also account for only up to 8% of the total class abundance of the MS profile (Figure 5C). This places a significant focus on changes within the subspecies presented here. The analysis of the acyl-chains shows a significant difference in the abundance of C22 sn2-FA between CPMS and RRMS (Appendix A). A notable feature is seen in the RRMS variant exhibiting an increase in 3 PC O- subspecies (Figure 4B).

In contrast, considering the PE class, we found changes affecting about 80% of the class’s total mass (Figure 5C). About 2/3 of these changes are attributed to the RRMS course. Moreover, PE ranks in the top three classes with the most significant changes across all comparisons (Appendix A). Thus, changes in PE subspecies were highly weighted by the oPLS-DA to discriminate between healthy and RRMS (Figure 3B).

In the CPMS cohort, the ether-bound variant of PE (PE O-) held a unique position. This is based on the frequency (Figure 4E) and influence on the overall abundance (Figure 5C), compared to alterations observed in the RRMS cohort (Figure 4B and Figure 5C).

Despite the low abundance of PI compared to the other lipid classes (e.g., PC:PI (1:0.025)) reductions in PI subspecies were considered relevant in defining the MS pattern by the oPLS-DA (Appendix A) and RF (Figure 5A). This is due to the fact that about 80% of the total mass of PI were altered in the RRMS and CPMS cohorts (Figure 5C). A measure of 2/3 of this alteration was attributed to the RRMS cohort profile (Figure 5C).

DAG was the only lipid class that showed a significant increase in both MS variants compared to the healthy control (Figure 1B). DAG 16:0;0_18:1;0 was detectable exclusively in the two MS cohorts among all 120 participants, and DAG 18:1;0_18:1;0 was detectable in all RRMS and CPMS patients and only three healthy individuals. Thus, DAG 18:1;0_18:1;0 was significantly higher in both RRMS and CPMS compared to the corresponding controls. Therefore, these lipids were classified as excellent predictors by the machine learning algorithms (Figure 5A). The cohort-specific profiles also showed a reduction in DAG 18:2;0_20:3:0, DAG 18:1;0_18:2;0 and DAG 14:0;0_18:1;0 in the MS cohorts (Figure 4B–E), thus emphasizing that the DAG class is highly modulated in PwMS (Figure 5C).

### 2.5. Composition of Storage Lipids, in Particular TAG and CE Differs Amongst RRMS and CPMS Cohorts

In the cohort-specific lipid signatures of RRMS and CPMS, there was only one significantly altered lipid overlapping in each of the TAG and CE classes (Figure 4B,C). In the CPMS cohort, significantly altered CE species account for about 60% of the total CE lipid mass, whereas in RRMS cohort the corresponding significantly altered CE species represent around 14% of the total CE lipid mass (Figure 5C). Contrarily, for TAGs in the RRMS course, the 20 significantly altered lipids (Figure 4B) comprise about 5–10% of the total mass whereas the corresponding altered TAGs in the CPMS course represent less than 1% of the total mass (Figure 5C). However, as the RF (Figure 5A) and oPLS-DA (Figure 3B,E and Appendix A) demonstrate, the lipid species of both lipid classes have low predictability as diagnostic biomarkers.

In the Cer class, 95% of the total mass undergoes alteration in PwMS. Of these changes, approximately two-thirds can be ascribed to the CPMS course. This significant shift is driven by four ceramide species–Cer 40:0;2, Cer 40:1;2, Cer 40:2;2, and Cer 42:1;2–showing a decline.

After considering the course-related lipid profiles, we proceeded to explore differences in disease activity. For this purpose, a subgroup analysis by activity within the RRMS and CPMS cohorts was performed. Using the aforementioned selection procedure, we were able to detect significantly altered lipids (Figure 6A). Three of these lipids were remarkable not only for achieving statistical significance but also for their substantial log_2_FC—upper left quadrant of Figure 6A. Comparing all significantly altered lipids of this analysis with the significantly altered lipids from the RRMS and CPMS specific lipid profiles (Figure 6B), 20 lipids emerged as potential activity markers (Figure 6C).

## 3. Discussion

In this study, we used quantitative shotgun lipidomics to identify a panel of altered lipids in 60 patients with MS compared to age-matched control groups. Based on additional clinical data as well as multivariate data analysis and machine learning, we were able to isolate distinct lipid profiles associated with either a relapsing-remitting or a chronic progressive course as well as a shared lipidomic pattern related to the diagnosis of MS in general, independent of additional clinical features. The common pattern mainly consists of a reduction in a subset of the PC, PC O- and PE lipid species as well as an increase in the DAG species (Figure 4C,D). This pattern was validated by an RF machine learning algorithm, demonstrating high sensitivity and specificity across all 120 participants, as illustrated by an AUC of 1 in the ROC curve (Appendix A). Prior studies in various smaller cohorts, including a cohort of monozygotic twins discordant for MS, support our findings [2,4,15]. Our diagnostic precision was driven by the DAG species 16:0;0_18:1;0, present only in PwMS (Figure 5A). It is therefore inevitable that our RF model is prone to overfitting; however, DAG species in general appear to be of relevance in the plasma lipid composition of PwMS, as demonstrated in previous studies as well [2]. Regarding the course-specific profiles, our results highlight that PE alterations are predominantly associated with RRMS (Figure 3A,B), whereas PE O- alterations are more characteristic of CPMS (Figure 3D,E). Furthermore, as demonstrated by the CCV we established, alterations in the CE and Cer subspecies compositions are present predominantly in the CPMS cohort (Figure 5C). In addition, we provide insight into possible lipidomic changes in patients with a higher clinical activity (Figure 6A–C).

### 3.1. DAG Elevation as a Hallmark in MS Diagnosis

In general, a significant increase in the total DAG abundance (Figure 1B) and in DAG subspecies is a remarkable feature of PwMS and proves to be a parameter independent not only of the course of the disease but also of possible confounders such as age and BMI (Figure 3A,D and Figure 5A). A previous study, predominantly involving patients with a relapsing-remitting course, also found elevated DAG subspecies to be correlated with MS [2]. However, primary progressive MS seems to be an exception due to a contrary correlation that was observed [15]. This is in line with the fact that our CPMS cohort predominantly consisted of secondary progressive MS patients. Interestingly, in a small comparison of newly diagnosed PwMS with a healthy cohort, the pattern of increased DAG was also found in the cerebrospinal fluid [22,23]. This suggests that the alterations in DAG presented in our dataset might arise from pathological processes within the CNS. The synthesis of DAG is significantly influenced by phosphatidic acid phosphatase, which catalyzes the dephosphorylation of phosphatidic acid to diacylglycerol [24] and diacylglycerol O-acyltransferases by the transfer of an acyl group to diacylglycerol [25]. The regulation of dephosphorylation, which itself is regulated by mTOR [26] and in cross talk with TREM2 [27], as well as the transfer of the acyl-chain [28] have been shown to be altered in MS and might be one potential link between the lipid metabolism and pathways of immunoregulation in PwMS. In addition, it can be hypothesized that the increased myelin turnover and release during demyelinating processes in the CNS [29,30] is reflected by lipid debris from myelin being transferred to storage lipids such as DAG and subsequently released into the plasma.

### 3.2. Impaired Ether-Bound Lipids in MS Progression

We were able to identify an extensive subset of PE lipid species mainly altered in patients with RRMS, whereas a higher proportion of PE O- species were reduced in patients with a chronic progressive course of the disease (Figure 3A,B,D,E and Figure 5C). Ether lipids are a unique class of glycerophospholipids. An alkyl chain is attached to the sn-1 position by an ether bond. Plasmalogens have an additional cis double bond next to the ether linkage. The mass spectroscopy setting used in our study does not allow the specification of these double bonds. However, due to their high portion in the total phospholipid mass (about 20% in human cell membranes), it is reasonable to assume that the ether lipids measured represent plasmalogens in a relevant fraction, particularly in the CNS [31,32]. A reduction in the PE O- in plasma is already known in other neurodegenerative diseases such as Alzheimer’s disease [33]. However, the causality is still unknown, as demonstrated by unsuccessful intervention studies [34]. Recently, the new assignment of the rate-determining enzyme (FAR1) of ether lipid biosynthesis to lipid droplets (LD) [32] links the synthesis of ether lipids to the remyelination of MS plaques [27]. Focusing on the signaling pathways in remyelination, the impairment of the liver X receptor (LXR) accompanied by alterations in ether lipid synthesis is prominent [30,35], thus pointing toward a potential role of ether lipids in processes related to remyelination.

So far, limited data on the comparison of normal-appearing white matter tissue of patients with primary progressive or secondary progressive MS has shown a reduction in putative plasmalogens, not further differentiated, in secondary progressive MS [3].

An analysis of the non-ether bound variants of PE species showed a significant reduction considering the total lipid class in both MS courses compared to the controls (Figure 5C), consistent with previous studies [2]. In contrast to the ether-bound variant, these changes in PE subspecies can be attributed to patients with a relapsing-remittent disease course as shown by the CCV (Figure 5C). The significantly altered lipids represent the typical distribution of polyunsaturated fatty acids in the sn2 position. Considering the cleavage products of PE, previous experiments with EAE mice demonstrated the requirement of LPE (1-18:1) for the activation of pathogenic Th17 cells [36]. In our study, we can attribute the significant change of this specific LPE to the RRMS course (Figure 4B). However, this change was measured under therapy with predominantly highly active substances that are known to reduce Th17 cell numbers [37]. Thus, the therapeutic effect as a potential confounder should be considered in this regard.

### 3.3. Cholesterol Esters and Triacylglycerols as Indicators of Clinical Features

Interestingly, we were also able to demonstrate differences in the composition of CE and TAG species in correlation with different courses of the disease. These lipid classes are relevant as storage lipids in the human lipid metabolism. We detected a significant reduction in CE species in the CPMS cohort (Figure 4E and Figure 5C) in contrast to a prominent reduction in TAG species in the RRMS course (Figure 4B and Figure 5C). Aware of the potential influence of aging on the lipidomic changes observed in CPMS, we have implemented strategies to mitigate this. First, we used an age-matched control group (OND) without a significant age difference. In addition, we applied a correlation filter for aging effects, mostly selecting PE O- species (Appendix A). Changes in approximately 3% of the total amount of CE and Cer species, which are significantly altered in the CPMS cohort, have been associated with aging by Hornburg et al. [38]. The changes shown here are well above 3% of the total mass and thus suggest a CPMS-specific profile (Figure 5C).

Cholesterol metabolism has been shown to be of importance for various aspects of MS pathophysiology leading from its influence on neuroinflammation in the demyelination [39,40], remyelination [29,30] as well as the neurodegeneration associated with the disease [41]. In the context of LDs, storage lipids, such as TAG, appear to be modulating neuroinflammation and, in particular, microglial activity [39,40],which is thought to be a pivotal factor in the pathophysiological processes involved in the chronic progression and diffuse white matter pathology in PwMS. It is therefore of interest that lipid profile changes regarding CE and TAG species are associated with distinct clinical features and courses of the disease, making these lipid patterns (Figure 4B,E) valuable potential candidates as biomarkers to assess individual states of the disease.

While previous studies have emphasized significant changes in Cer and SM due to their relative high proportion in myelin [2,15,42], our analysis reveals only minor changes in comparison to age-matched controls (Figure 1B and Figure 5C). This suggests that a fraction of Cer and SM alterations might be related to aging effects. However, it should be noted that the reported changes in Cer exceed the age-related changes described in the literature [38].

### 3.4. MS Lipidomic Patterns—A Unique Signature with Overlaps to Other Diseases

Despite our identification of disease-course-specific lipid profiles in the plasma, the origin of these patterns remains elusive. From our observations, several hypotheses regarding this issue can be drawn. Changes in the plasma lipid composition in patients may be associated with the activation of the peripheral and/or central immune system or might reflect alterations in CNS lipid composition due to the damage of lipid-rich myelin sheaths. Intriguingly, it is plausible that both processes coexist. Given the uncertain origins of the lipidomic profiles, examining the pattern identified in this study alongside those of plasma lipidomics with a similar depth of resolution of other diseases could provide clarity. Comparing our findings to altered plasma lipid profiles in a degenerative neurological disease like Parkinson’s disease reveals a slightly different pattern, i.e., a decrease in TAG as well as an increase in PE and DAG [43]. Motoneuron diseases like amyotrophic lateral sclerosis and primary lateral sclerosis show, in contrast to our findings, an increase in CE subspecies (e.g., CE 24:5 and CE 24:2), Cer subspecies, DAG and TAG; commonly a reduction in PC and PC O- is found [21]. This reinforces our assumption that the patterns described in Figure 4B–E are distinct from other neurodegenerative processes.

Considering inflammatory states, a transient decrease in lipid subspecies abundances consisting of TAG, LPC and PE O- was found to correlate with an elevation in c-reactive protein and neutrophils during respiratory infections [38]. In the case of inflammatory diseases such as rheumatoid arthritis, various reductions in lysophosphatidyls, PC, and PC O- have been found. Most strikingly, no increase in DAG or changes in PE, PE O-, CE or Cer were mentioned [44,45].

Similar to these findings, we observed a reduction in the TAG, PE O-, PC, and PC O- subspecies. Yet, in addition, we noted a decrease in the CE, Cer, PC, PC O-, and PI subspecies, accompanied by an increase in DAG.

In contrast to our profiles, Lauber et al. determined a decrease in DAG and an increase in PI, PC, and TAG in patients with a high risk for cardiovascular events [46]. However, DAG 18:1;0_18:3;0, considered a marker for individuals at highest risk for cardiovascular diseases, is structurally related to DAG 16:0;0_18:1;0, which showed the highest predictivity in our study (Figure 5A). Moreover, the reductions in CE, PE and PE O- (Figure 4B–E) were also associated with cardiovascular diseases. These comparisons reveal a unique signature of MS and its specific courses, with overlap to other inflammatory and degenerative diseases.

We have identified a distinct lipid signature associated with the diagnosis of PwMS. Furthermore, we were able to distinguish different lipidomic profiles indicating distinct clinical disease courses. The main characteristic of these profiles is a reduction in most lipid subspecies. Only DAG shows a significant increase regardless of the disease course. Additionally, our study points toward the potential of lipidomic signatures in assessing the disease activity of inflammatory demyelinating CNS diseases. Our study has several limitations. Although our investigation of 120 participants is more comprehensive than prior lipidomic studies in MS [4,15,42], it is still below the typical size for biomarker studies [38,47]. Moreover, only 13 patients of our study were classified as active, indicating that they had clinical activity in the 12 months prior to sampling. Therefore, the lipids shown in Figure 6 provide only a first insight calling for more comprehensive and possibly prospective studies. Finally, the DMT was not taken into account as a confounding factor due to the limited sample size.

Further studies recruiting larger cohorts are needed to confirm and to specify the promising results of our study in the context of previous results [2,4,15]. Plasma as well as CSF lipid profiles might pave way to a better understanding of individual states of the disease in MS, thus facilitating tailored therapeutic approaches in patients suffering from the disease.

## 4. Materials and Methods

### 4.1. Study Cohorts

Patients were recruited during a routine follow-up visit at the Centre for Multiple Sclerosis at University Medical Centre Göttingen. Sixty of the patients who fulfilled the 2017 McDonald criteria for MS were enrolled, with participants evenly divided into two groups: 30 with a course of RRMS and 30 with a course of chronic progressive MS. Chronic progression was defined by the accumulation of neurological deficits that occur continuously and mainly independent of relapse activity. In the context of this study, ‘activity’ refers to the occurrence of clinical relapses in the 12 months prior to sampling [48]. Patients with a predominantly progressive variant, both with and without additional relapses, constitute the chronic progressive MS (CPMS) cohort of our study. Activity was further used to stratify the two MS cohorts into active MS and inactive MS subgroups.

Current disease-modifying therapies (DMTs) are listed in the Appendix A). Disease duration denotes the interval from diagnosis to the moment of plasma collection. Concurrently with sample acquisition, the Expanded Disability Status Scale (EDSS) was documented. Collection took place between March and June 2021. Controls consisted of healthy students or employees of the clinic. These healthy controls were not diagnosed with any medical conditions and were not taking any medications at the time of the study. To ensure age comparability, we also included 30 patients from our neurological outpatient clinic. These patients attended routine appointments during the same period and had neurological diseases without central nervous system involvement (OND).

In accordance with epidemiology, the CPMS cohort was older, with an average age of 56 years compared to 28 years in the RRMS cohort, as well as experiencing a significantly longer disease duration (Table 1 and Appendix A).

The active MS subgroup was predominantly composed of patients from the RRMS cohort (85%) and reported shorter disease durations compared with the inactive subgroup (Table 2 and Appendix A).

### 4.2. Ethics Approval and Informed Consent Statement

This study was conducted in accordance with the Declaration of Helsinki, and approved by the Ethics Committee of the University Medical Center Göttingen under application number 09/10/10, with the final endorsement provided on 6 March 2014. All participating patients were fully informed about the nature and extent of the data collected and provided written consent for the pseudonymized processing and publication of their data.

### 4.3. Lipidomics

Blood was drawn in the morning before therapeutic interventions and plasma was stored at −80 °C within 30 min of collection.

Prior to analysis, samples were thawed at 4 °C. The shotgun nano-electrospray high-resolution Orbitrap mass spectrometry was performed by Lipotype GmbH (Dresden, Germany) as described in [18]. A total of 670 lipids, characterized according to the LipidMaps nomenclature, were identified in this study.

### 4.4. Statistics

We developed an automated statistical platform for the analysis of our lipidomic datasets. As a first step, we performed data imputation and normalization procedures. In addition, filtering based on age or BMI correlation was applied. We employed both traditional statistical methods, such as the Welch’s *t*-test, as well as unsupervised and supervised multivariate data analyses, such as PCA and oPLS-DA. To validate patterns from the volcano plot and oPLS-DA and assess predictive capability, we used a machine learning random forest algorithm. A detailed description of the analysis can be found in the Appendix A).

## Figures and Tables

**Figure 1 ijms-25-02483-f001:**
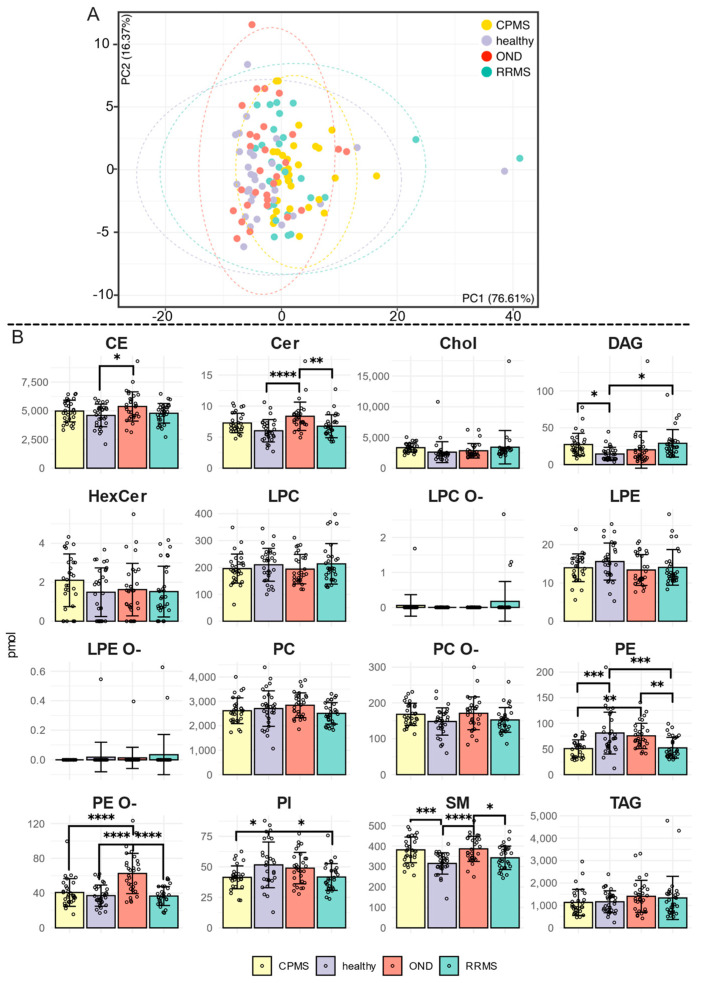
(**A**) Principal Component Analysis (PCA) plot illustrating the lipid subspecies distribution from 120 participants. The variance explained by PC1 is 76.61% and PC2 is 13.79%. Ellipses represent the 95% confidence intervals for each cohort. (**B**) Bar chart illustrating the concentration [pmol] with SD for the 16 lipid classes. Statistical significance was determined by a one-way ANOVA, followed by the TUKEY-HSD: * = *p*-value < 0.05; ** = *p*-value < 0.01; *** = *p*-value < 0.001; **** = *p*-value < 0.0001 (Appendix A).

**Figure 2 ijms-25-02483-f002:**
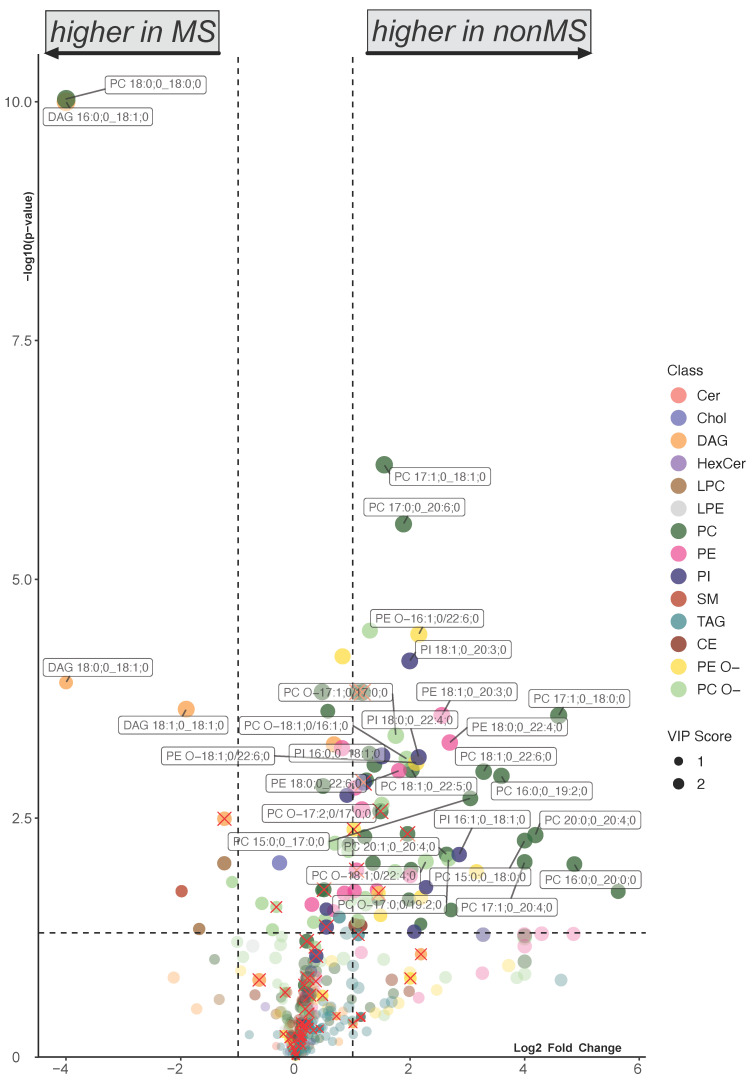
Volcano plot showing the differences in lipid subspecies in subjects without MS (healthy and OND) vs. subjects with MS (RRMS and CPMS). Colors represent lipid class classification. The size of the dots is determined by the VIP score from the comparative oPLS-DA (Appendix A). Lipids marked with red × have a high correlation with age and BMI (Appendix A). The horizontal dashed line indicates a *p*-value of 0.05 in the Welch’s *t*-test. The vertical dashed line indicates a log_2_ fold change of 1. Only lipids with a *p*-value < 0.01 and a log_2_ fold change > 1.5 and no relevant correlation with age and BMI were annotated.

**Figure 3 ijms-25-02483-f003:**
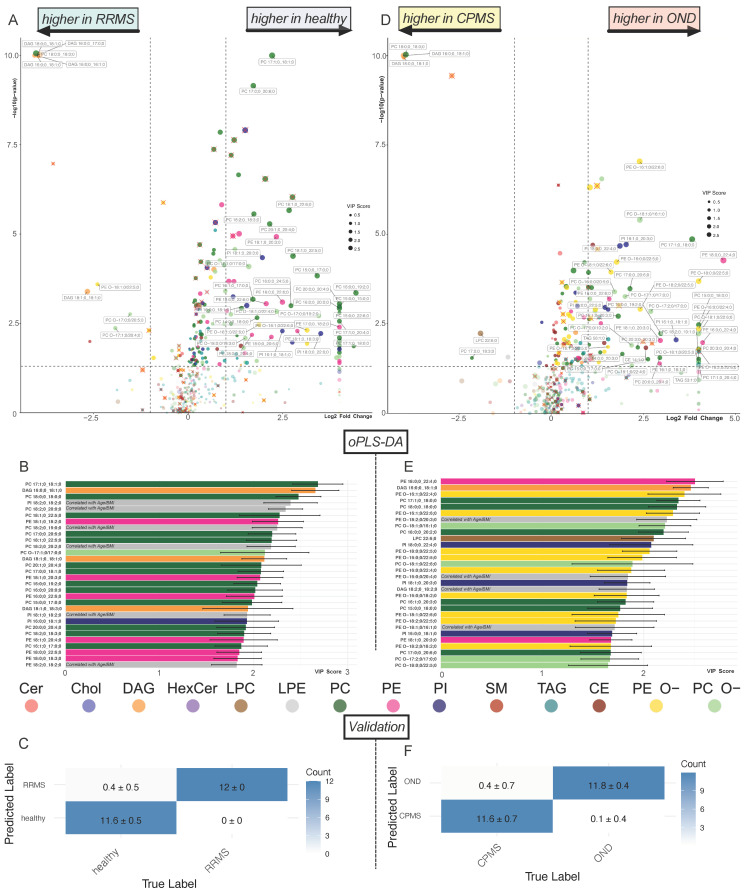
Volcano plot showing the differences in lipid subspecies in healthy subjects vs. patients with RRMS (**A**) and subjects with OND vs. CPMS (**D**). Colors represent lipid class classification. Lipids marked with red × have a high correlation with age and BMI (Appendix A). The horizontal dashed line indicates a *p*-value of 0.05 in the Welch’s *t*-test. The vertical dashed line indicates a log_2_ fold change of 1. Only lipids with a *p*-value < 0.01 and a log_2_ fold change > 1.5 and no relevant correlation with age and BMI were annotated. The size of the dots is determined by the VIP score from the oPLS-DA (**B**,**E**). (**B**) Bar chart displaying the lipids with the top 30 VIP scores with SD of oPLS-DA healthy vs. RRMS (Appendix A). (**E**) Bar chart displaying the lipids with the top 30 VIP scores with SD of oPLS-DA OND vs. CPMS (Appendix A). Colors in (**B**,**E**) represent lipid class classification. Lipids with a high correlation with age and BMI are greyed out and annotated. Confusion matrix of the oPLS-DA of healthy vs. RRMS (**C**) and OND vs. CPMS (**F**) on the 40% hold out test dataset (testing data) ensuring the predictive capability of the important lipids provided by the oPLS-DA.

**Figure 4 ijms-25-02483-f004:**
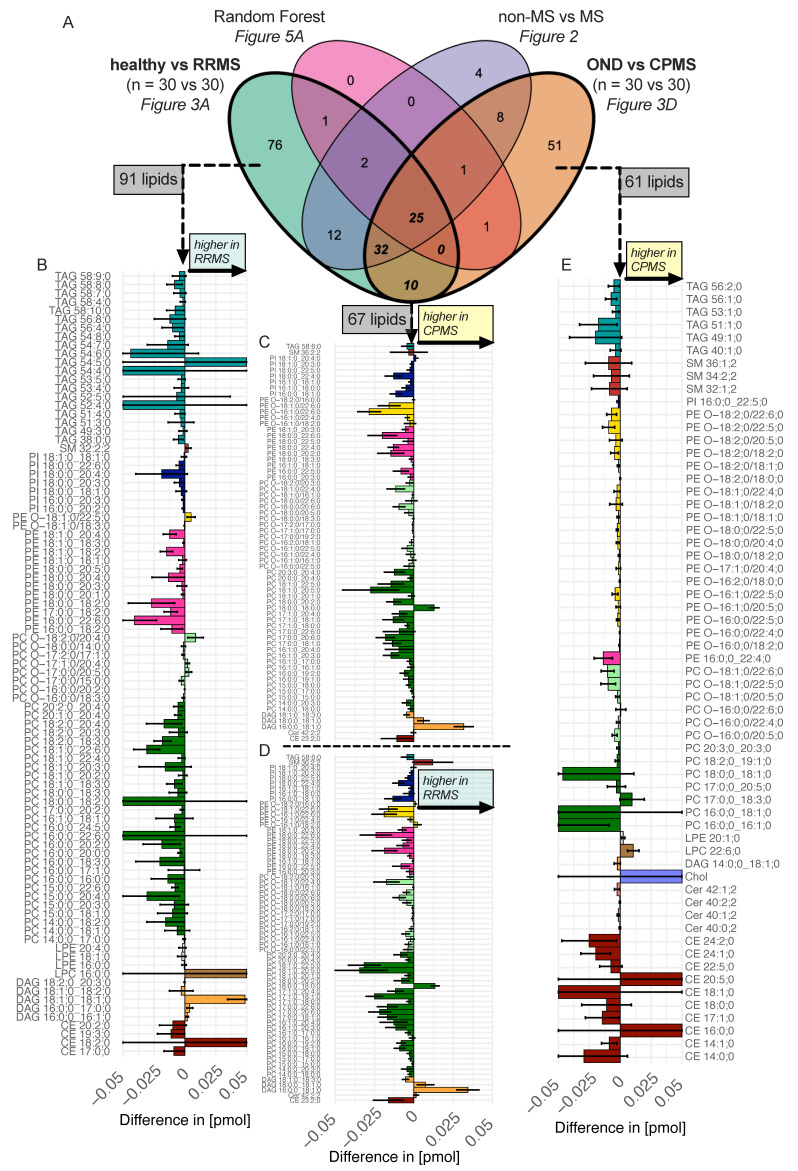
Cohort-Specific Significantly Altered Lipids. (**A**) Venn-Diagram illustrating the overlaps in the significantly altered lipids across three comparative analyses using oPLS-DA and Volcano: “healthy vs. RRMS (Figure 3A)”, “non-MS vs. MS (Figure 2)” and “OND vs. CPMS (Figure 3D)” and the Random Forest machine learning classification “non-MS vs. MS (Figure 5A)”. By analyzing the overlaps presented in the Venn diagram, it becomes evident which lipid alterations are specific to individual cohorts. This overlap-based approach enables a precise identification of cohort-specific lipidomic signatures. The individual lipids of these signatures are shown in (**B**–**E**). (**B**) Represents the signature for the RRMS cohort. (**C**,**D**) Illustrate the common overlap of lipids that are altered regardless of MS progression. (**E**) Represents the signature for the CPMS cohort. The horizontal bar plots detail the absolute difference amount within the range of −0.05 to 0.05 pmol (mean differences beyond this range truncated) along with their 95% confidence intervals.

**Figure 5 ijms-25-02483-f005:**
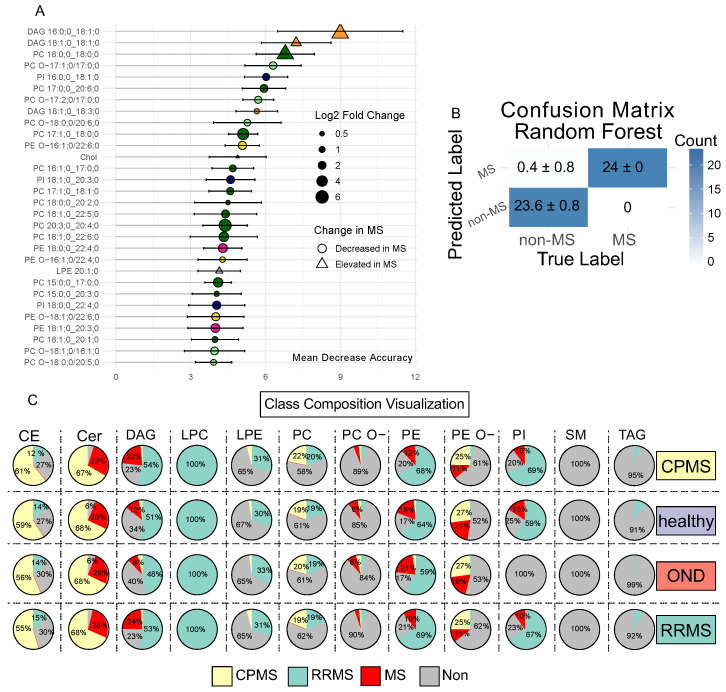
Random forest and class composition visualization. (**A**) Shows the top 30 lipids considered most discriminative by the random forest model when comparing (healthy and OND vs. RRMS and CPMS), ranked by their mean decrease in accuracy. Notably, DAG 16:0;0_18:1;0 emerges as the most discriminative lipid. The predictive ability of the model was measured using a separate test dataset, as shown in (**B**). To determine whether the marked absolute differences in lipid subspecies shown in Figure 4B–E significantly influence the overall composition of the associated lipid class, we calculated the total class quantity in each cohort and visualized the significantly altered lipids based on their cohort specificity determined by Figure 4A. This approach is illustrated as a class composition visualization in (**C**).

**Figure 6 ijms-25-02483-f006:**
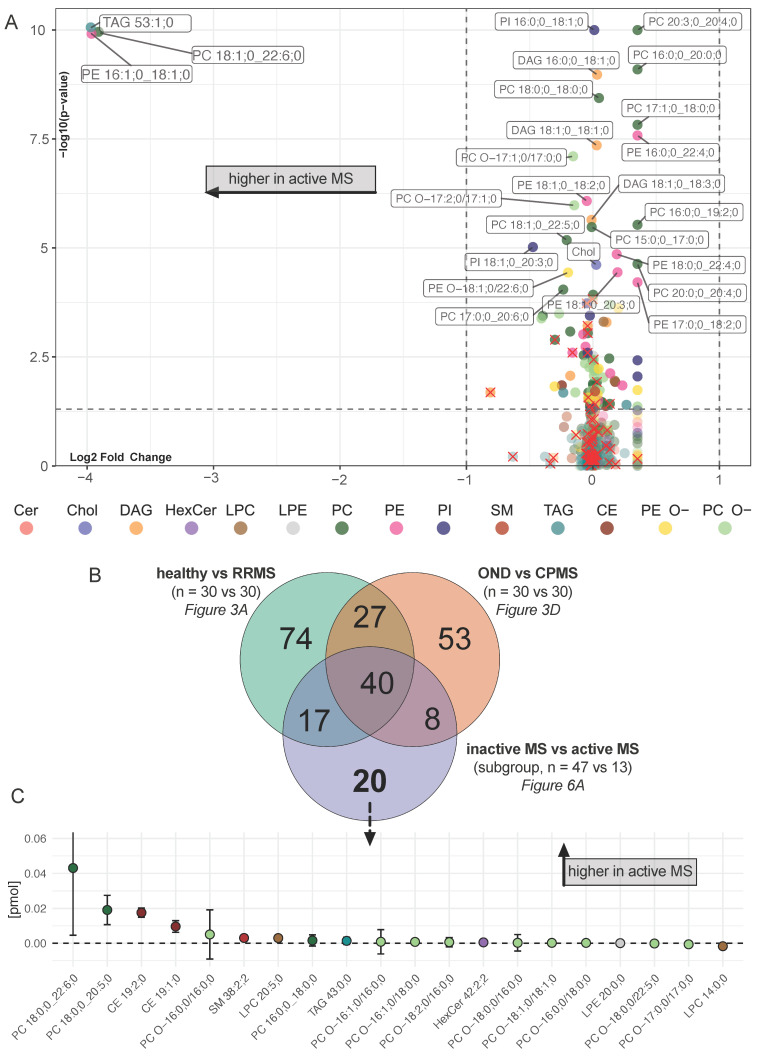
Differences in lipid compositions in inactive MS vs. active MS. (**A**) Volcano plot showing the differences in lipid subspecies in patients with inactive MS vs. patients with active MS. Colors represent lipid class classification. Lipids marked with red × have a high correlation with age and BMI (Appendix A). The horizontal dashed line indicates a *p*-value of 0.05 using the Welch’s *t*-test. The vertical dashed line indicates a log_2_ fold change of 1. Only lipids with a *p*-value < 0.0001 and no relevant correlation with age and BMI were annotated; (**B**) comparing the overlaps of significantly altered lipids from the prior comparative analyses of “healthy vs. RRMS (Figure 3A)” and “OND vs. CPMS (Figure 3D)” to “inactive MS vs. active MS (**A**)”. We identified 20 lipids that are significantly altered only in patients with active MS. (**C**) Illustrates these significantly altered lipids by presenting the differences in pmol with SD.

**Table 1 ijms-25-02483-t001:** Cohort Description—Demographics and Clinical Variables.

Parameter	Healthy	RRMS	CPMS	OND
n	30	30	30	30
Age [mean ± SD]	23.6 ± 2.3	27.5 ± 4.3	55.7 ± 13	52.8 ± 10.3
BMI [mean ± SD]	22.8 ± 2.7	25.6 ± 7.5	24.8 ± 4.9	27.9 ± 6.1
Sex [M/F]	12/18	10/20	11/19	10/20
EDSS at Sampling [mean ± SD]	-	2.5 ± 1.6	5.3 ± 1.2	-
Disease duration, yrs. [mean ± SD]	-	3.6 ± 3.5	18.7 ± 12.4	-

F = female; M = male; SD = standard deviation.

**Table 2 ijms-25-02483-t002:** Subgroup Description—Demographics and Clinical Variables.

Parameter	Active MS	Inactive MS
n	13	47
Age [mean ± SD]	30.1 ± 10.7	44.8 ± 17.3
BMI [mean ± SD]	24.9 ± 6.3	25.3 ± 6.3
Sex [M/F]	5/8	16/31
EDSS at Sampling [mean ± SD]	2.9 ± 1.8	4.2 ± 1.9
Disease duration, yrs. [mean ± SD]	3.3 ± 5	13.3 ± 12.2

F = female; M = male; SD = standard deviation.

## Data Availability

Data not included in this article are available in an anonymized form and can be requested by contacting the corresponding author. In accordance with the recommendations of the International Lipidomics Society, the reporting checklist of the Lipidomics Standards Initiative [51] is included in the Appendix A (Supplementary LSI-Checklist).

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
