# Peer review of "Plasma Lipidomic Profiling Using Mass Spectrometry for Multiple Sclerosis Diagnosis and Disease Activity Stratification (LipidMS)"

_ijms, 2024, doi:10.3390/ijms25052483_

Round 1

Reviewer 1 Report

Comments and Suggestions for Authors

The research focuses on Plasma Lipidomic Profiling by Mass Spectrometry in MS to draw the potential clinical contribution for disease monitoring or personalized medicine. However there are many junk or mass signals in Figure 2, figure 5 BC, and Figure 6 A, which must be improved and should be friendly to the reader.

2. You only divided into active, inactive MS and health control, but you did not provide the BMI information of enrolled cases, and their age distribution, which should be challenged, due to lipid profile may be changed by daily food contents, please provide these informations and discuss this part, you know obesity is a chronic inflammtory  status may influence your information.

Reviewer 2 Report

Comments and Suggestions for Authors

This is a very nice work trying to elucidate more the pathogenesis of multiple sclerosis. 

It is not clear to me if there could be any direct implication into clinical practice (I think no) and what would be the further direction of this research to make it more practical.

I see that there are some remarks about CSF but I would appreciate some more info - do you expect cholesterol to be the same in circulating blood and CSF? What do you you expect is the influence of blood brain barrier on lipid levels in these two compartments? Would it be not more informatory to start with CSF examination? 

Most studies in MS with cholestorol lowering agents did not show effect. Can you comment on it? (You mention only Alzheimer´s.) As brain is full of lipids, one need to know the relationship between lowering of blood lipids and well-being of CNS cells first. What is the relationship? Do we have enough information? (not to make more harm than benefit)

Round 2

Reviewer 2 Report

Comments and Suggestions for Authors

The authors responded to all my questions with adequate knowledge and experience in the field.